# Single-Shot 3D Topography of Transmissive and Reflective Samples with a Dual-Mode Telecentric-Based Digital Holographic Microscope

**DOI:** 10.3390/s22103793

**Published:** 2022-05-17

**Authors:** Ana Doblas, Charity Hayes-Rounds, Rohan Isaac, Felio Perez

**Affiliations:** 1Department of Electrical and Computer Engineering, The University of Memphis, Memphis, TN 38152, USA; cwhite23@memphis.edu; 2FedEx Institute of Technology, The University of Memphis, Memphis, TN 38152, USA; rmisaac@memphis.edu; 3Material Science Laboratory, Integrated Microscopy Center, The University of Memphis, Memphis, TN 38152, USA; fperez@memphis.edu

**Keywords:** common-path interferometry, digital holographic microscopy, Fresnel biprism

## Abstract

Common path DHM systems are the most robust DHM systems as they are based on self-interference and are thus less prone to external fluctuations. A common issue amongst these DHM systems is that the two replicas of the sample’s information overlay due to self-interference, making them only suitable for imaging sparse samples. This overlay has restricted the use of common-path DHM systems in material science. The overlay can be overcome by limiting the sample’s field of view to occupy only half of the imaging field of view or by using an optical spatial filter. In this work, we have implemented optical spatial filtering in a common-path DHM system using a Fresnel biprism. We have analyzed the optimal pinhole size by evaluating the frequency content of the reconstructed phase images of a star target. We have also measured the accuracy of the system and the sensitivity to noise for different pinhole sizes. Finally, we have proposed the first dual-mode common-path DHM system using a Fresnel biprism. The performance of the dual-model DHM system has been evaluated experimentally using transmissive and reflective microscopic samples.

## 1. Introduction

Digital holographic microscopy (DHM) provides accurate three-dimensional (3D) measurements of microscopic unstained samples with high lateral resolution (up to 200 nm) in a non-invasive way. DHM measures optical path length changes, enabling quantitative phase imaging (QPI) with minimum sample preparation. Among the multiple advantages of DHM systems, we can highlight its applicability to dynamic samples, enabling time-lapse imaging from milliseconds to multiple days, monitoring dynamic processes in biomedical and material sciences. For instance, DHM systems have been used to monitor mitosis [1,2,3] and cell culture quality in biomedical applications, providing an accurate quantification of cell culture parameters [4]. In addition, QPI-DHM provides biophysical cell properties, such as cell volume, refractive index, and cell mass, which are parameters related to multiple cellular processes [5,6]. Further, DHM has been widely utilized as a blood analyzer, identifying blood cells infected by malaria [7,8,9], screening people with diabetes [10], and sickle cell anemia [11,12] based on their red blood cells, as well as other inherited anemias [13]. Recently, O’ Connor et al. presented a compact and field-portable, 3D-printed shearing digital holographic microscope to screen red blood cells infected with COVID-19 [14]. In addition, QPI-DHM has allowed the identification and characterization of cancer cells and therapies [15,16,17]. In material science, the most common applications of DHM are related to the evaluation of MEMS [18,19,20,21] and the analysis of static and dynamic surface topography [22,23,24]. Dynamic DHM imaging has been used to characterize sample changes due to external mechanical and electromagnetic forces and/or thermodynamic variables (e.g., pressure, temperature) [25]. These DHM systems are considered 4D (3D + time) optical profilometers, providing widefield measurements without scanning and with sub-nanometric axial accuracy.

DHM systems are optical interferometers in which a digital camera records the interference pattern between two coherent waves. Traditional DHM systems follow a Mach–Zehnder or Michelson configuration based on the sample type. For instance, unstained biological samples present low optical scattering and reflectivity, making them almost transparent in bright-field imaging. Therefore, unstained biological samples require DHM systems operating in their transmission mode (e.g., Mach–Zehnder setup). In contrast, reflective samples such as MEMS and other semiconductor components require Michelson-based DHM systems (e.g., reflection-based DHM systems). Regardless of their optical configuration, both traditional DHM systems rely on the fact that the two interfering waves travel different optical paths, making them more sensitive to temporal fluctuations. Common-path DHM systems rely on the principle that both interfering waves travel nearly the same optical path, making them the most stable and robust DHM systems. These common-path DHM systems require the use of an optical element such as a diffraction grating [26,27], a Wollaston prism [28], a lateral shear plate [29,30], a beamsplitter [31,32,33], or a Fresnel biprism [34,35,36,37] to generate a self-interference pattern. The hallmarks of common-path systems are their temporal stability and compactness, making them suitable for implementation as an external module in current commercial microscopes. However, common-path DHM systems can only provide quantitative phase images of microscopic samples with low spatial density despite these advantages. This limitation is related to the fact that the self-interference pattern is generated by two replicas of the object image. Consequently, common-path DHM systems have mostly been reported for sparse biological samples. To avoid any undesired superposition between these two object replicas, some researchers have restricted the usable image field of view by inserting the microscopic sample within half of the illuminated field of view [35]. Another approach to removing any potential overlay that does not restrict the image field of view is the optical spatial filtering of one of the interference beams [27,32,33,34]. Finally, later in 2021, Weng et al. [38] proposed a common-path off-axis DHM system using the polarization-based Senarmont prism. The spatial overlay between both beams can be removed by rotating the Senarmont prism. However, this approach cannot effectively eliminate the spatial overlay for highly dense samples, restricting its use for low and medium dense samples. In this work, we propose a compact common-path DHM system for reconstructing the 3D topography of transmissive and reflective samples (i.e., dual-mode DHM system) using a single recorded image (i.e., single-shot technique). The proposed common-path DHM system is based on a telecentric microscopic imaging system, a Fresnel biprism (FB), and a 4f imaging system with a spatial filter (i.e., pinhole) located at its Fourier plane. The advantage of our design is its compactness, making it suitable for an external module adaptable to any commercial microscope to obtain accurate diffraction-limited phase images regardless of the sample type.

## 2. Dual-Mode Digital Holographic Microscopy Using a Fresnel Biprism

### 2.1. Compact FB-Based DHM System

Over the last decade, FBs have been used in common-path DHM to create the self-interference pattern, enabling low-cost, compact, and robust (less prompt to temporal fluctuations) DHM systems. Figure 1 illustrates one of the most simplified versions of the FB-based common-path DHM system in which the FB is inserted in the image space of the DHM imaging system [37]. The microscopic sample is illuminated by a uniform plane wave in this configuration. The complex amplitude distribution scattered by the sample, o(⋅), is then imaged through a conventional optical microscope. This microscope is a telecentric imaging system composed of an infinity-corrected 40× Nikon microscope objective (MO) lens with a numerical aperture (NA) of 0.75 and a tube lens (TL) of focal length 200 mm. Telecentric-based imaging systems in QPI-DHM provide intrinsically linear shift-invariant systems, enabling accurate phase measurements without posteriori computational methods [39,40] and diffraction-limited phase images (i.e., the resolution of the reconstructed phase images is limited by diffraction theory) [41]. A CMOS sensor (5472 × 3648 px^2^ with 2.4-µm square pixel size) is placed at the image plane of the optical microscope (i.e., the back focal plane of the TL lens), allowing the acquisition of in-focus images. Mathematically, the complex amplitude distribution of the object distribution at the image plane, *u_IP_*(*x*,*y*), is given by
(1)$$u_{IP}\left(x,y\right)\propto \frac{1}{M^{2}}o\left(\frac{x}{M},\frac{y}{M}\right){⊗}_{2}P\left(\frac{x}{\lambda f_{\mathrm{TL}}},\frac{y}{\lambda f_{\mathrm{TL}}}\right),$$where ${⊗}_{2}$ denotes the 2D convolution operator, (*x*, *y*) are the transverse spatial coordinates, *λ* is the illumination wavelength, and *M* = −*f*_TL_/*f*_MO_ is the lateral magnification of the imaging system with *f*_MO_ and *f*_TL_ being the focal lengths of the MO and TL lenses, respectively. Since we have used the focal length of the TL recommended by the MO manufacturer, the lateral magnification of the DHM imaging system coincides with the one displayed in the MO lens. In Equation (1), P(⋅) is the 2D Fourier transform of the amplitude transmittance of the pupil distribution, p(⋅). Some irrelevant factors have been omitted in Equation (1).

Since the Fresnel biprism is inserted between the TL and the sensor plane, the biprism produces two separated replicas of the object image: $u_{IP}\left(x-s\tan\delta ,y\right)$ and $u_{IP}\left(x+s\tan\delta ,y\right)$, where *s* is the axial distance between the biprism’s vertex and the camera plane, and *δ* is the refringence angle of the biprism. For simplicity, we have considered that the biprism’s central edge is aligned with the *y*-axis, separating the object distributions along the *x*-axis. The separation between these object replicas, 2stanδ, depends linearly on the distance *s*. The further the biprism is to the camera sensor, the bigger the separation is between the object replicas. The interferential fringes generated by the self-interference of the object replicas are confined in a rhombus-shaped region (see Figure 1 in [37]). The maximum fringes’ field of view (FOV), which theoretically is equal to half the lateral extension of the biprism (*L*), assuming that the biprism is the optical limiting element (i.e., FoVmax = *L*/2 = 20/2 = 10 mm), is found when the biprism’s vertex is located at $s_{\max}=L/(4\tan\delta )$ from the sensor’s camera. Experimentally, the maximum fringes’ FOV has been measured to be 9.684 mm. No interferential fringes are found for axial distances (*s*) equal to or higher than $2s_{\max}=L/(2\tan\delta )$. As the phase information in DHM can only be reconstructed if the sample information is encoded within the interferential fringes, the axial position of the FB should be such that it produces the highest fringe’s FOV. This position would provide reconstructed phase images with the highest FOV without moving the microscopic sample laterally. Therefore, in this scenario, the lateral separation distance between the two object replicas is separated by *L*/2. The irradiance distribution of this self-interference pattern, commonly called a hologram, is
(2)$$h\left(x,y\right)={\left|u_{IP}\left(x-L/4,y\right)+u_{IP}\left(x+L/4,y\right)\right|}^{2},$$where |·|^2^ represents the square modulus. Whereas the reference beam is a uniform plane wave in dual-path DHM systems (e.g., DHM systems based on Mach–Zehnder interferometer), the reference beam is a replica of the complex object distribution in common-path DHM systems. Therefore, common-path DHM systems are restricted to sparse samples (i.e., low-density microscopic samples in which the sample information is spatially dispersed). Figure 1b,c illustrate the overlay problem between the object replicas in the common-path FB-based DHM system. Let us consider that the region of interest in a microscopic sample has a lateral extension of Δ_x_, producing a magnified image of *M*·Δ_x_ through the DHM imaging system. The separation between both replicas is governed by the axial position of the FB, being a maximum of (*L*/2). Only when the size of the object images (*M*·Δ_x_) is smaller than the separation between the two replicas, *M*·Δ_x_ ≤ *L*/2, is there no overlay between the replicas, and one of the replicated object waves is considered as a uniform wave, enabling DHM imaging. If the size of the microscopic sample (Δ_x_) and the separation of the replicas (*L*/2 to ensure the maximum fringes’ FOV) are fixed parameters, the overlay between the two object replicas can be diminished by reducing the lateral magnification of the imaging system. The lower the lateral magnification of the optical microscope, the smaller the overlay between the two replicas. The reduction of the lateral magnification can be achieved by changing the TL for one with a lower focal length and/or changing the MO lens for one with a lower lateral magnification. Whereas the first case is not practical, the reduction of the lateral magnification by changing the MO lens typically involves the reduction of the numerical aperture of the MO lens, providing microscopic images with low resolution. Therefore, this solution is unsuitable for high-resolution quantitative phase imaging (i.e., images with finer resolvable details).

### 2.2. FB-Based DHM System for Spatially-Dense Microscopic Samples

Figure 2 shows the optical configuration of the proposed common-path DHM system using an FB and a spatial filtering system. For simplicity, we have neglected the sample and the MO lens. The spatial filtering system is composed of a 4*f* imaging system (i.e., two converging lenses, L1 and L2 lenses, arranged in an afocal configuration) and a pinhole set at the Fourier plane of the 4*f* imaging system (i.e., back focal plane of the L1 lens and front focal plane of the L2 lens). This spatial filtering system eliminates the medium and high spatial frequencies of one of the object images generated by the FB to create the interference pattern between one of the object images and a uniform reference wave. In addition, this proposed FB-based DHM system still operates in the telecentric regime. In this configuration, in-focus DHM holograms with the highest fringes’ FOV are located at the back focal plane of the L2 lens (i.e., new image plane) since the new and native IPs in the proposed DHM system are conjugated planes (i.e., IP and IP’ are set at the front focal plane of L1 and back focal plane of L2, respectively). Since the L1 and L2 lenses have the same focal length (i.e., *f*_L1_ = *f*_L2_ = 125 mm), the lateral magnification of the DHM imaging system, *M* = −*f*_TL_/*f*_MO_, and the modulation period of the interferential fringes, *p*_m_ = 1/*u*_m_ = *λ*/[2(*n* − 1) tan(*δ*)] where *n* is the refractive index of the biprism, remain the same values. A USAF target from the QPM target (Benchmark Technologies) has been used to evaluate the spatial resolution of the DHM system, shown in Figure 2a. Figure 2b shows the reconstructed phase image provided by our system using a pinhole size equal to 30 μm. The reconstructed phase image has been obtained using the approach proposed by Castaneda et al. [42]. The smallest resolvable feature in the USAF target is the 9-4 element, corresponding to a distance equal to 691 nm. The percentage difference between the experimental value and the theoretical expectation (*λ*/NA = 532 nm/0.75 ≈ 709 nm) is approximately 2.5%, verifying that the proposed DHM system still operates at the diffraction limit, reconstructing high-resolution microscopic information.

The key element of the proposed DHM system in Figure 2 is the insertion of the optical spatial filtering (i.e., the pinhole). Pinholes of diameters 25 μm and 50 μm have previously been reported in common-path DHM systems [32,34]. In addition, Bhaduri et al. provided the limit of the maximum diameter of the pinhole for the diffraction phase microscopy [27], which is similar to a common-path DHM system with a diffraction grating. To determine the optimal diameter of the pinhole, we have evaluated the performance of our DHM system by imaging a star pattern from the QPM target. The area of the star’s FOV is 0.0662 mm^2^ (=π *r*^2^ with a radius *r* = 145.2 μm), and the smallest resolvable detail is up to 274 μm (corresponding to the 10-6 element in the USAF target). To ease the changing of the pinhole, we have used a pinhole wheel with 16 pinholes of a diameter from 25 μm to 2 mm (PHWM16, Thorlabs). The pinhole wheel was mounted onto a translation stage for its proper alignment. Appendix A shows how to align the pinhole of 25-μm.

Figure 3 shows the recorded holograms for different pinhole diameters. The exposure time of the CMOS camera was adjusted for each hologram, ensuring that the dynamic range of the hologram was maximum. The spatially filtered object replica (i.e., the uniform reference wave) starts containing sample information for a pinhole size equal to or higher than 200 μm. Holograms recorded with a pinhole size equal to 90 μm and below look pretty alike. The reduction of the pinhole size also affects the intensity of the reference beam, leading to a reference wave with less power. Theoretically, the contrast of an interference pattern is only maximum when the power of the two interfering beams is the same [43]. We increased the camera’s exposure time to compensate for the reduction of the fringes’ contrast due to the diameter of the pinhole. The exposure time was increased from 416 µs for the 2-mm pinhole to 830 µs for the 30-µm pinhole. Despite this increase, the highest exposure time still allows us to track living cells and analyze real-time dynamics. We have evaluated the fringes’ contrast of the hologram by measuring the maximum and minimum values of the fringes. The fringes’ contrast was estimated along the direction marked by the red line in Figure 3a. By definition, the fringes’ contrast (*C*) is given by the ratio between the difference of the maximum and minimum values and their sum, *C* = [max − min]/[max + min]. The mean and standard deviation of the fringes’ contrast is shown in Figure 3a for each experimental hologram. We do not observe a reduction in the fringes’ contrast within the experimental error due to the increase in the camera’s exposure time. Figure 3b shows the reconstructed phase images of the star target using the reconstruction approach described in [42]. As shown in Figure 3b, the reconstructed phase images for a 2-mm pinhole size present the overlay between the two replicas of the star target. Each replica of the star target has a different phase value, even though these replicas come from the same target. This phase difference between the replica images is due to a phase difference of π introduced by the FB between each interfering wave. Although the reconstructed phase image for a pinhole size of 200 μm does not show the second replica, one can realize that its phase image is distorted by some phase nuisances coming from the second replica, restricting the usable field of view. The reconstructed phase image for the 90-μm pinhole is also slightly ruined by phase nuisances, marked by yellow ovals in Figure 3b.

In contrast, a pinhole size of 50 μm or below should provide reconstructed phase images with minimum phase distortions. For both pinhole sizes, the measured phase values (*ϕ*) have been converted into thickness via *t* = [*ϕ λ*]/[2π(*n*_g_ − *n*_m_)], assuming *n*_g_ = 1.52, *n*_m_ = 1, and *λ* = 532 nm. We have measured the thickness value (mean ± standard deviation) of the star target for the 30-μm and 50-μm pinhole to be equal to 398 ± 41 nm and 355 ± 9 nm, respectively. For both pinhole sizes, the measured thickness agrees with the one provided by the manufacturer, *t* = 350 nm, within the experimental errors. Although the difference between the reconstructed phase images is minimal, the frequency distribution of these phase images differs. Figure 4 compares the Fourier spectrum of the reconstructed phase images for both pinholes. Panel (a) shows a composite RGB image to display the frequencies with a different magnitude in the spectrum of the reconstructed phase images. The gray-colored frequencies for the 50-µm pinhole refer to the frequencies with the same magnitude as the 30-µm pinhole. However, the green-colored frequencies refer to the ones only present for the 50-µm pinhole. To further highlight the difference in the Fourier spectrum, we have plotted the radial sum power spectrum versus the radial frequency. The radial sum power spectrum is the sum of all possible directional power spectra. Figure 4b illustrates the increase in the frequency content as a direct consequence of the size of the pinhole. Even though a pinhole size of 50 μm creates a uniform reference beam without phase nuisances in its reconstructed phase image, its spectrum distribution has been altered.

## 3. Dual-Mode Telecentric-Based Digital Holographic Microscope

The optical configuration of the proposed common-path DHM using an FB and a spatial filtering system has been upgraded to image both transmissive and reflective microscopic samples. Figure 5 shows the configuration of the dual-model common-path DHM using the biprism-based QPI module. The Appendix A provides the list of components and the alignment protocol to construct this system. The dual-mode DHM system has two distinct illumination systems for imaging transmissive and reflective microscopic samples. The illumination sources in both imaging modalities are the same; a low-power collimated laser diode module (CPS532, Thorlabs) with a center wavelength of 532 nm. The illumination beam emerging from the laser diode has a circular diameter of approximately 3.5 mm. Note that the illumination beam in the reflection-based imaging modality is demagnified by a factor of *f*_MO_/*f*_TL_, which is equal to 40× for *f*_MO_ = 200 mm/40 = 5 mm, and *f*_TL_ = 200 mm. Due to this shrinkage, the illumination beam in the reflection-based illumination system has been enlarged by a factor of 10× using an achromatic Galilean beam expander (GBE10-A, Thorlabs). Therefore, reflective samples are illuminated by a uniform plane wave with a diameter equal to 875 µm (i.e., field of view of the illumination beam is 875 µm). The light scattered by transmissive and reflective specimens is collected by the same imaging system comprising the infinity-corrected 40×/0.75 NA Nikon MO lens and the TL of a focal length of 200 mm. The MO and TL lenses still operate in the telecentric regime to optically compensate for the spherical aberrations introduced by the MO lens [39,40].

The proposed FB-based QPI-DHM module for spatially dense microscopic samples has been slightly modified to provide a compact QPI-DHM module, enhancing its utilization in commercial microscopes. We have reduced the length of the FB-based QPI-DHM module by using L1 and L2 lenses with a shorter focal length (i.e., 75 mm instead of 125 mm), leading to a reduction of 1.67×. Since a pinhole with a diameter of 50 μm introduces additional spatial frequencies in the spectrum of the reconstructed phase images (Figure 4b), the QPI-DHM module has a 30-µm pinhole (P30K, Thorlabs). This element is mounted onto a translational stage (ST1XY-S/M, Thorlabs) to facilitate its alignment. The two-point sources generated by the FB at the Fourier plane (i.e., back focal plane of L1 lens = front focal plane of L2 lens) are separated by a lateral distance equal to 6.54 mm (i.e., 2 *λ f*_L1_ *u*_m_ for *λ* = 0.532 µm, *f*_L1_ = 75 mm, and *u*_m_ = 0.082 µm^−1^). To generate the uniform reference beam, only one of these point sources should be spatially filtered by the 30-µm pinhole. Therefore, we drilled a hole of 4-mm diameter, approximately at 6.5 mm in the outer part of the mount from the 30-µm pinhole. We ensured that diffraction-limited phase images were reconstructed by drilling a hole with a diameter large enough that the frequency content of the object beam was not filtered. The interference pattern between both object and reference beams was again recorded by the Basler sensor (5472 × 3648 px^2^, 2.4-μm^2^ pixel size). The performance of the FB-based QPI-DHM module was validated by imaging a transmissive USAF target from the QPM target, verifying that the system’s resolution of 691 nm was achieved and high-resolution phase images were reconstructed. The performance of the FB-based QPI-DHM module was also demonstrated by imaging normal (healthy) unstained human red blood cells (RBCs) on a microscopic slide from Carolina Biological Supply Company (#item C25222, https://www.carolina.com/histology-microscope-slides/readi-stain-human-blood-smear-unstained-microscope-slides-set-of-15/C25222.pr, accessed on 9 April 2022). Unstained RBCs have been widely imaged in DHM, enabling the detection of malaria [7,9] and screening of diabetes [10], COVID-19 [14], and sickle cell anemia [11,12] as well as other inherited anemias [13]. Figure 5c shows the 3D topographical view of the unwrapped reconstructed thickness map. We have used the algorithm described in [44] to unwrap the reconstructed phase images. The RBC’s thicknesses (*t*) were obtained from the unwrapped phase values (*ϕ*) via *t* = [*ϕ λ*]/[2π(*n*_s_ − *n*_m_)] where *λ* = 532 nm is the illumination wavelength, and *n*_s_ = 1.406 and *n*_m_ = 1 are the refractive indices of the RBCs [45] and the surrounding media, respectively. We have measured an average optical thickness of 0.4 µm, which is 2× smaller than the value reported in [46]. This reduction may be related to the dehydration of the stored RBCs experienced over time versus the fresh RBCs reported in [46].

Finally, the performance of the FB-based QPI-DHM module for imaging reflective samples has been evaluated by imaging a high-resolution USAF target (HIGHRES-2, Newport) and an array of squares from the logo of Photomask Portal. A 40-nm film was grown on both samples via the DC-sputtering technique using an EMS 550 coater (Electron Microscopy Sciences Hatfield, PA, USA). The layer was grown at room temperature and 10 mbar of argon with high purity (99.99%). The sputter current was 40 mA, and the base pressure of the system was 10^−3^ mbar. Each sample was sputtered at a 20 nm/min deposition rate from a Gold/Palladium (80/20) target material with 99.99% purity. The target material was located 35 mm above the microscopic samples. Figure 6 shows the thickness maps of the USAF target using a commercial profilometer (Profilm3D, Filmetrics) and the proposed common-path DHM operating in reflection mode. For the case of our off-axis common-path DHM operating in reflection and telecentric mode (Figure 5b), the thickness map has been converted from the phase measurements (*ϕ*) via *t* = [*ϕ λ*]/[4π] with *λ* = 532 nm. We have measured the thickness value (mean ± standard deviation) of the high-resolution target to be equal to 66 ± 11 nm. This thickness agrees with the one provided by the profilometer, *t* = 74.59 nm (average value), within the experimental errors. The lateral resolution power of the commercial profilometer is not enough to resolve groups 8 and 9 of the high-resolution USAF target. Whereas the minimum resolvable feature in Figure 6a is the 7-3 element (i.e., a lateral resolution of 3.10 µm), we can discriminate objects laterally separated up to 691 nm (i.e., 9-4 element in Figure 6b) along the horizontal and vertical axes. The lateral resolution is kept constant for both imaging modalities, enabling high-resolution 3D topography maps in both the transmission and reflection modes. The main result of the single-shot DHM operating in reflection mode must be its capability to reproduce the 3D topography of the imaged sample; this result is shown in Figure 6b. This panel shows the good quality of the 3D-topography map obtained for the reflective USAF resolution test.

In the final experiment, an array of squares from the logo of the Photomask Portal of a silicon MEMs wafer was imaged with the proposed FB-based QPI-DHM system operating in the reflection mode and a telecentric regimen. The thickness maps are presented in Figure 7 for both the commercial profilometer (top row) and our common-path QPI-DHM system (bottom row). The commercial profilometer does not provide enough resolution to image this sample; one can barely discriminate the squares from panels (a) and (b). Nonetheless, our FB-based QPI-DHM system provides high-contrast phase and thickness measurements. The structural features of the squares are: (1) a lateral period of (9.78 ± 0.48) µm; and (2) a separation between the squares of (3.6 ± 0.42) µm; and (3) thickness of (96 ± 9) nm. These results show that a reflective FB-based QPI-DHM system allows high-resolution measurements of the structural details in a sample.

## 4. Conclusions

In summary, we have presented a common-path DHM system using an FB operating in transmission and reflection modes that provide high-resolution 3D topographic maps of transmissive and reflective samples. The imaging system operates in a telecentric regime, providing reconstructed phase and thickness maps with minima phase aberrations. A complete protocol description to implement such a system in an open-setup configuration is provided in the Appendix A. The proposed common-path DHM system is suitable for transmissive and reflective samples, not restricting its utilization for highly dense samples. This feature is enabled since we have used an additional 4f imaging system with a spatial filter at its Fourier plane. The proposed system has been validated by imaging transmissive and reflective commercial targets. To the best of our knowledge, this is the first dual-mode common-path QPI-DHM system using an FB, enabling a high resolution and accurate quantitative phase and thickness measurements without limiting the sample’s size. Future work will focus on validating the polarization-sensitivity capability of the dual-mode QPI-DHM system using the FB in both imaging modalities. In addition, we will also investigate the limits of the sample’s reflectivity, analyzing the accuracy of the reconstructed phase images for different samples’ reflection ratios.

## Figures and Tables

**Figure 1 sensors-22-03793-f001:**
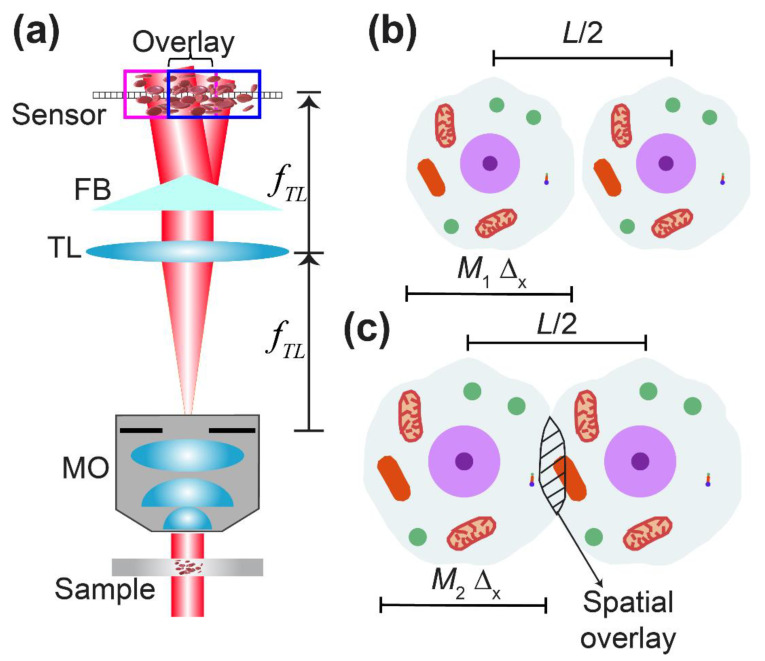
Simplified FB-based common-path digital holographic microscopy: (**a**) optical configuration; (**b**) illustration of the spatial overlay between the replicas of the object images, restricting the use of the FB-based common-path DHM system to sparse samples; (**c**) illustration of spatial overlay circled in red for dense samples.

**Figure 2 sensors-22-03793-f002:**
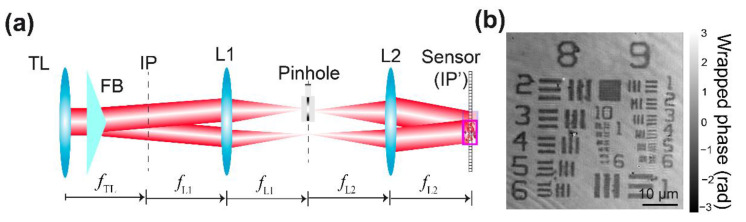
(**a**) The optical configuration of the proposed common-path DHM using a Fresnel biprism and a spatial filtering system to create the interference pattern between the complex amplitude distribution scattered by object and imaged by the system and a uniform reference wave. (**b**) Reconstructed phase image of a USAF target from the QPM target.

**Figure 3 sensors-22-03793-f003:**
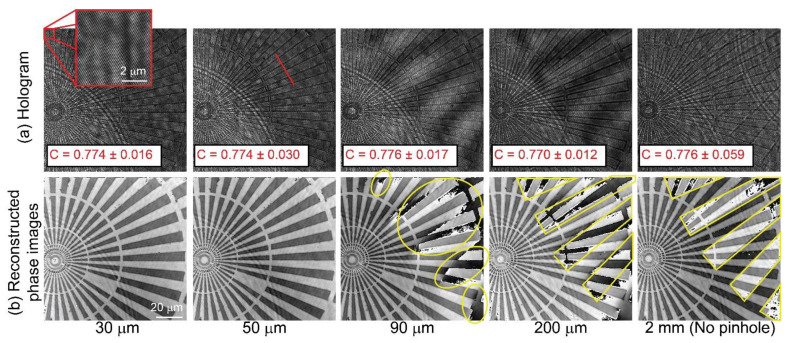
Evaluation of the pinhole size in the FB-based DHM system: (**a**,**b**) Experimental holograms and reconstructed phase images of the star pattern for different pinhole sizes. The inset in panel (**a**) shows the interference pattern created by the FB. The value of the fringes’ contrast estimated along the red direction (red font) is shown in panel (**a**).

**Figure 4 sensors-22-03793-f004:**
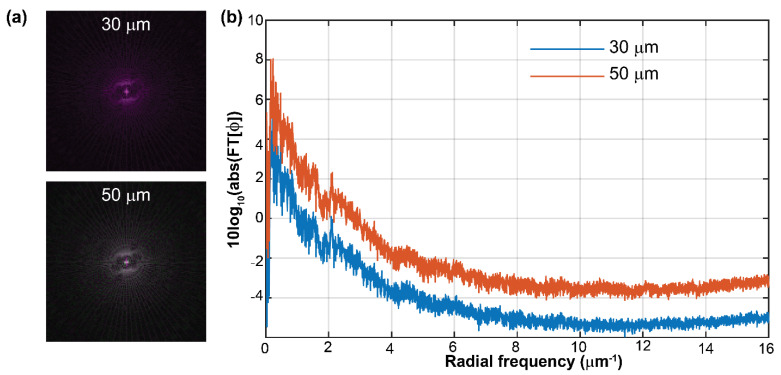
Comparison of the Fourier spectrum of the reconstructed phase images for a pinhole size of 30 and 50 μm in the FB-based DHM system. (**a**) The 2D Fourier transform of the reconstructed phase images for the pinholes of 30 and 50 μm. (**b**) Power spectrum distribution versus the radial frequency for both pinholes.

**Figure 5 sensors-22-03793-f005:**
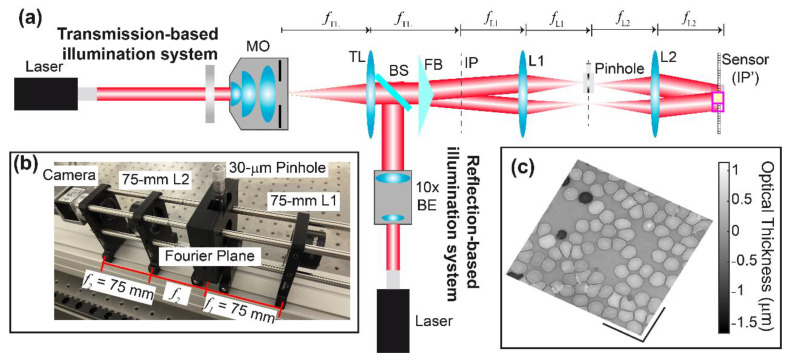
(**a**) The optical configuration of the dual-mode common-path digital holographic microscopy using a Fresnel biprism for quantitative phase imaging of transmissive and reflective samples. (**b**) Experimental setup of the compact QPI module with a spatial filtering system to generate a uniform reference beam. (**c**) Three-dimensional distribution of the optical thickness of human RBCs using the transmission-based illumination system. The scale bar is 25 µm.

**Figure 6 sensors-22-03793-f006:**
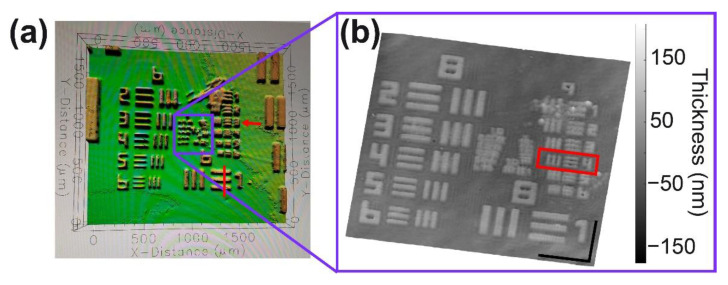
Reflective high-resolution USAF 1951 target imaged with (**a**) a profilometer and (**b**) the proposed common-path DHM operating in reflection mode. The scale bars in panel (**b**) are 14.4 µm. The red rectangle in panel (**b**) refers to the smallest resolvable element of the USAF target.

**Figure 7 sensors-22-03793-f007:**
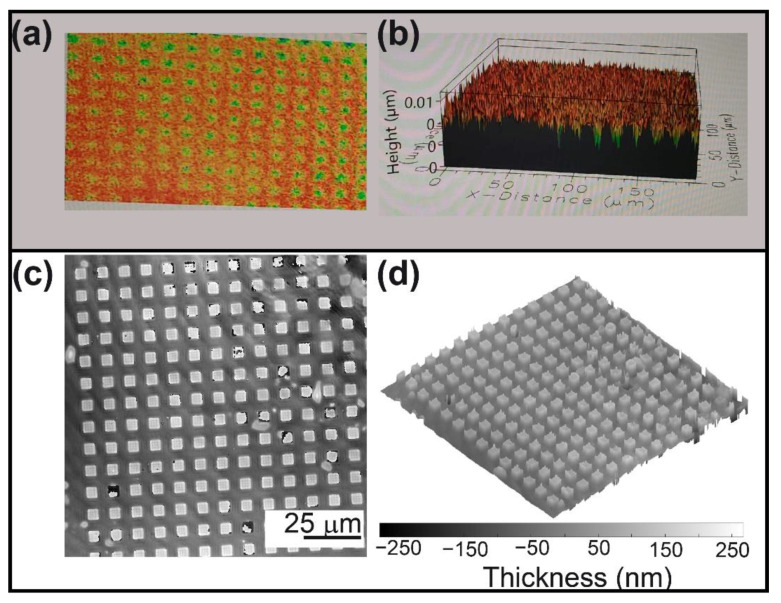
Reflective array printed in the logo of Photomask Portal. Thickness maps results obtained with (**a**,**b**) a profilometer and (**c**,**d**) the proposed common-path DHM operating in reflection mode.

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
