# Peer review of "Single-Shot 3D Topography of Transmissive and Reflective Samples with a Dual-Mode Telecentric-Based Digital Holographic Microscope"

_sensors, 2022, doi:10.3390/s22103793_

Round 1

Reviewer 1 Report

The authors implemented optical spatial filtering in a common-path quantitative phase imaging using a Fresnel biprism and a pinhole, in both reflective and transmission modes.

My remarks:
1. Please show magnified images of selected locations in the holograms so that the readers can see the fringe patterns.
2. Please add imaging of transparent samples in transmission mode, such as biological cells in a dish under large magnification.
3. You missed citing closely related works that use the same principle, for example:
Optics Express 21, 5701, 2013. Advances in Optics and Photonics 6, 57, 2014

Author Response

Attached is the response for the reviewer. 

Reviewer 2 Report

The authors have presented an interesting optical configuration with a Fresnel biprism which is better than the author's previous works. The main advancement in this method is the introduction of a pinhole to filter out most of the information of one of the object waves. This is a simple yet smart method. The authors have demonstrated it in both reflection as well as transmission configurations which is good. Can this method be applied to samples with continuous phase variation. In the first figure, the authors showed a sample that is a biological specimen but in the demonstration only standard samples have been used. At least can you comment on what to expect with samples with continuous phase variation. Overall the manuscript is well written with good quality figures. One minor concern is that the convolution operator. Is it necessary to add 2 to indicate 2d convolution? Might be better to just say that the symbol means 2d convolution. It appears odd in that equation. But it is upto the authors. There are some typos even in the abstract for interference and minor grammar errors.

Author Response

Attached is the response of the reviewer.

Reviewer 3 Report

In this manuscript, the authors presented a common-path digital holographic microscope (DHM) for 3D topographic applications in both transmission mode and reflection mode. They introduced the method to generate a pair of object images by using a Fresnel biprism. A 4f imaging system with a pinhole has been used to filter high spatial frequencies of one object image to create the reference beam. They demonstrated the experiment results of a USAF 1951 target and an array of squares sample. In my opinion, the authors gave a feasible design of a high robust DHM, and it would be helpful for scientific community. Therefore, I suggest that the manuscript would be accepted if the authors could make some further improvements. Here are my comments:
1.In line 108, what is the meaning of “o” in equation (1)?
2.What is the actual size of the highest fringe’s FOV in the proposed system?
3.In line 224, the authors mentioned that they did not find any reduction of the fringes’ contrast because they increased the exposure time of the camera. Please explain the reason and provide the value of exposure time.
4.In line 244, there are no green addition spatial frequencies in Fig.3(b).
5.A pinhole has been used in the 4f imaging system. The size of the pinhole is important for the elimination of the high special frequencies of one object wave, besides the overall dimension of the pinhole should not obstruct the high special frequencies of another object wave. The authors need provide the structure details of the pinhole, and discuss the potential influence for high-resolution applications. 
6.Are there any limits of the sample’s reflection ratio? Some commercial profilometers are robust in all surface situations from 0.05% to 100% reflectivity.

Author Response

Attached is the response to the reviewer's comment

Round 2

Reviewer 1 Report

The authors addressed my previous remarks. However, per my request, they added a biological cell image, and this image is too noisy.  Can you improve the results? Are you taking a hologram without the sample and subtract the phase extracted from this hologram to compensate for the spatial noise? Also, please add a colorbar to show how high the cell is.

Author Response

Once again, we appreciate the reviewer taking the time to read our revised manuscript. A detailed reply to the reviewer’s comment is provided in the attached document.
